# Considerations on Temperature Dependent Effective Diffusion and Permeability of Natural Clays

**DOI:** 10.3390/ma14174942

**Published:** 2021-08-30

**Authors:** Florian Wesenauer, Christian Jordan, Mudassar Azam, Michael Harasek, Franz Winter

**Affiliations:** 1Institute of Chemical, Environmental and Bioscience Engineering, Vienna University of Technology (TU Wien), 1060 Vienna, Austria; michael.harasek@tuwien.ac.at (M.H.); franz.winter@tuwien.ac.at (F.W.); 2Institute of Chemical Engineering & Technology, University of the Punjab, Lahore 54000, Pakistan; mudassar.icet@pu.edu.pk

**Keywords:** clay, brick, carbonates, diffusion, permeability, temperature, Wicke–Kallenbach cell, mean transport-pore model, Darcy’s law, Graham’s law

## Abstract

A series of porous clay samples prepared at different pretreatment temperatures have been tested in a diffusion chamber. Diffusivity and permeability were examined in a temperature range from ambient to 900 °C. Gaseous mixtures of O_2_, CO_2_, and N_2_ have been applied, as these species are the relevant gases in the context of clay brick firing and similar thermochemical processes. Diffusive transport characteristics have been determined by means of the mean transport-pore model, and permeability has been evaluated by Darcy’s law. CO_2_ diffusivity increased strongly with temperature, whereas O_2_ diffusion was limited to a certain level. It is proposed that one should consider CO_2_ surface diffusion in order to explain this phenomenon. The diffusion model was expanded and surface diffusion was included in the model equation. The results of the model fit reflected the important role of incorporated carbonates of the clay foundation in gas-phase (molecular or Knudsen) diffusivity. CO_2_ surface diffusion was observed to exhibit similar coefficients for two different investigated clays, and is therefore indicated as a property of natural clays. Permeability showed a progressive rise with temperature, in line with related literature.

## 1. Introduction

Diffusion in porous media is a widely investigated field in materials science and reaction kinetics research. In many heterogeneous reactions, such as catalytic reactions [1], iron ore induration [2], and adsorption processes [3], and also in geological processes [4], diffusive and convective transport in the porous solid phase both play a prominent role for gaseous reactants fluxes and consequently for overall reaction rates. In order to apply the fundamental equations of heat and mass transport problems, i.e., mass and energy balances, a comprehensive description of material properties is required. It is concluded that transport coefficients for a given heterogeneous system should be determined in the relevant range of process conditions [1]. However, due to a lack of experimental data, diffusion coefficients are frequently assumed to be a simple function of molecular diffusion coefficients, or are determined together with chemical kinetic parameters from experimental data [3,5]. The permeability of construction materials is usually evaluated by water vapor permeability according to standardized procedures [6,7], whatis not directly comparable with gas permeability. Experimental data for the diffusivity of porous materials are available in the literature, but usually only related to materials at ambient temperature. For example, Hou et al. investigated catalyst pellets at elevated temperatures and pressures by a diffusion cell method [1]. Horiuchi et al. studied surface diffusion effects up to 1073 K of CO_2_ on alumina modified with metal oxides such as CaO, MgO, etc. [8].

According to Guggenheim and Martin, “the term ‘clay’ refers to a naturally occurring material composed primarily of fine-grained minerals, which is generally plastic at appropriate water contents and will harden when dried or fired” [9]. It is further concluded that clay mainly consists of phyllosilicates, but inorganic compounds as well as organic matter are also usually found in natural clays. Phyllosilicates are structures formed by sheets of interconnected SiO_4_ tetrahedra, whereas Si can be substituted by Al or other atoms (Si^4+^ replaced by Al^3+^, Al^3+^ replaced by Mg^2+^ or Fe^2+^, etc.) [10]. Substitution requires the incorporation of positively charged cations for charge balance and implies a moderate layer charge, dependent on the specific mineral group (kaolinite, smectite, illite, etc.) [11]. Phyllosilicates are able to bind substantial amounts of water in the inter-layer space, whereas water is also absorbed in the bulk pores. Clay dehydration is the process of H_2_O release during heat exposition. Natural clay is generally a mixture of several minerals. It is not only characterized by the constitution of the dense phase, but also by a certain range of particle size of the bulk phase. Clay particles are commonly accepted to be in the colloidal range and specified as being smaller than 2 µm, and sometimes smaller than 4 µm [11]. Clays are often attributed to be low cost, mainly inorganic materials, which are available in many regions of the earth in almost unlimited amounts and exhibit manifold desirable physical and/or chemical properties. Clays have been deployed in industry and are the subject of recent research for various applications, among them use in ceramics, paper production, as insulation material and carriers of chemical substances, catalysts [10], adsorbents [12], filters [13], and so forth [14]. 

In the present study, the transport characteristics of clay samples have been examined in a broad range of temperatures by applying the mean transport-pore model (MTPM) and Darcy’s law. Samples were further characterized by mercury intrusion porosimetry of the clay mixtures and evaluation of particle size distribution and chemical composition of the raw clays. In order to investigate the influence of the degree of burnout on the diffusion behavior, samples were prepared with a temperature treatment of different strengths. Disc-shaped clay specimens were tested in a diffusion chamber test rig able to cover temperatures from ambient to 900 °C. Samples were provided by a clay brick manufactory and, hence, the results were assumed to exhibit a high suitability for practical application. 

## 2. Theoretical 

### 2.1. Gas-Phase Diffusion

The binary molecular diffusion coefficient, *D_ij_^m^*, is described by the well-known Chapman–Enskog equation (e.g., Gaskell [15]). Because molecular diffusion coefficients have been extensively investigated, empirical equations for molecular diffusion coefficients were taken here from Marrero and Mason [16]. Knudsen diffusion is closely related to the mean molecular speed, *v_i_^m^*, according to the kinetic theory of gases: (1)$$D_{i}^{K}=\frac{2}{3}r_{p}v_{m,i}=\frac{2}{3}r_{p}{\left(\frac{8RT}{\pi M_{i}}\right)}^{1/2}$$

Molecular diffusion coefficients are related to gas pairs, whereas, for the Knudsen diffusion regime, the wall-gas molecule collisions are relevant rather than the gas-gas interaction. Thus, the molar mass of only one concerned species determines the Knudsen diffusion coefficient. The mean free path length of a gas molecule is directly proportional to temperature:(2)$$\lambda_{i}=\frac{kT}{\sqrt{2}P\pi \sigma_{i}^{2}}$$
where *k* is the Boltzmann constant, *P* is the total pressure, and *σ_i_* is the kinetic diameter of gas species *i*. Table 1 lists values for gases concerned in the present work. *λ* is relevant for the characterization of the diffusion regime by the Knudsen number, relating *λ* with the mean pore radius ⟨*r*⟩: (3)$$Kn=\frac{\lambda_{i}}{2〈r〉}$$

The relation indicates the transition of gas-phase diffusion in porous media from molecular to the Knudsen regime, with values <<1 for wide molecular diffusion and >>1 for Knudsen diffusion as the dominant mechanism. *Kn* ~ 1 indicates a transition region. 

## 2.2. Surface Diffusion

While gas-phase diffusion is a strictly mechanistic process (as it is described by the kinetic gas theory), surface diffusion can be seen as a chemical complement of the overall phenomenon. Interaction of gaseous diffusants with the pore walls can promote surface flux in porous media. Surface diffusion is closely related to the heat of adsorption, as the interaction of a gas molecule with the pore wall is equivalent to a physical or chemical adsorption process [18]. A temperature dependency of the Arrhenius-type is commonly accepted: (4)$$D^{s}=D_{0}^{s}e^{-\frac{E_{a,s}}{RT}}$$
where *E_a,s_* is the difference of activation energy of surface diffusion and the heat of adsorption [19], referred to as *apparent* activation energy here. 

### 2.3. Diffusion Model

Soukop et al. [20] give a comprehensive description of local effective gas-phase diffusion by the mean transport-pore model (MTPM). A cylindrical transport pore with a mean diameter, ⟨*r*⟩, is introduced, together with *ψ = ε/τ*, the proportionality between effective and theoretically unhindered diffusion coefficients; ε is the porosity of the solid, while *τ* is the tortuosity factor, which considers the excess length of a real pore compared to the macroscopic diffusion direction, i.e., the non-straightness. Effective molecular and Knudsen diffusion coefficients in porous media are given by [20]: (5)$$D_{ij}^{m,e}=\frac{\varepsilon }{\tau }D_{ij}^{m}=\psi D_{ij}^{m}$$
(6)$$D_{i}^{K,e}=\psi 〈r〉K_{i}$$
where *K_i_ = 2/3(8RT/**π/M_i_)*^1/2^. Diffusive transport in porous media according to MTPM is described by a modified Maxwell–Stefan equation: (7)$$-c\frac{dy_{i}}{dx}=\frac{N_{i}^{d}}{D_{i}^{K,e}}+\sum_{\underset{j\neq i}{j=1}}\frac{y_{j}N_{i}^{d}-y_{i}N_{j}^{d}}{D_{ij}^{m,e}}$$

It is well-understood that gas-phase diffusion of a gas mixture is non-equimolar and described by Graham’s law, which is, for a binary mixture: (8)$$N_{j}^{d}/N_{i}^{d}=-\sqrt{M_{i}/M_{j}}$$

From Equation (7), gas-phase diffusive flux in a binary mixture of species *A* and *B* is derived as follows: (9)$$N_{A}^{d}=-c\frac{dy_{A}}{dx}{\left(\frac{1}{D_{A}^{K,e}}+\frac{1-\alpha y_{A}}{D_{AB}^{m,e}}\right)}^{-1}=-D_{AB}^{g}c\frac{dy_{A}}{dx}$$
where *α* = 1 − *(M_a_/M_b_)*^1/2^ and
$D_{AB}^{g}$ represents the effective gas-phase diffusion coefficient, which is the inverse of the sum of the reciprocals of molecular and Knudsen diffusion coefficients (1/*D^g^_AB_* ≈ 1/*D^K^* + 1/*D^m^*). From a mathematical point of view, the diffusion coefficients correspond to a conductance (e.g., electrical) treated as acting in series, yielding an overall diffusivity, which is generally smaller than each single contribution. This means that each molecule has not only to migrate by means of one or the other mechanism, but its unhindered diffusion is restricted by *both* gas-phase mechanisms.

To fit the experimental data, gas-phase and surface diffusion had to be considered for CO_2_ diffusion. It is reported that the surface diffusion coefficient can be highly dependent on the concentration of the adsorptive, i.e., the diffusant under certain conditions [19]. However, the molar flux of a diffusive transport phenomenon by *any* mechanism, regardless of some limitations also for surface diffusion, may be formulated by Fick’s first law [21]: (10)$$N_{i}^{d}=-\left(D_{i}^{g}+D_{i}^{s}\right)c\frac{\partial y_{i}}{\partial x}$$

Other authors give a more rigid formulation, taking into account the surface concentration dependency of the diffusant [19,22]. In contrast to the above considerations concerning the two different gas-phase mechanisms, gas-phase and surface diffusion are treated here in parallel. This seems justified by the concept of gas-phase diffusion as a kinetic and surface diffusion as a chemical contribution to total diffusive flux

Substituting Equations (4)–(6) resp. (9) into Equation (10), the following extended diffusion model for combined gas-phase and surface diffusion is obtained:(11)$$N_{A}^{d}=-\left[{\left(\frac{1}{\psi 〈r〉K_{A}}+\frac{1-\alpha y_{A}}{\psi D_{AB}^{m}}\right)}^{-1}+D_{0}^{s}e^{-\frac{E_{a,s}}{RT}}\right]c\frac{\partial y_{A}}{\partial x}$$

Integrating the above equation over the height, *L*, of the porous layer yields: (12)$$N_{A}^{d}=\frac{c}{L}[\frac{\psi D_{AB}^{m}}{\alpha }ln(\frac{\frac{D_{AB}^{m}}{〈r〉K_{A}}+1-\alpha \bar{y_{A}^{U}}}{\frac{D_{AB}^{m}}{〈r〉K_{A}}+1-\alpha \bar{y_{A}^{L}}})-(\bar{y_{A}^{U}}-\bar{y_{A}^{L}})D_{0}^{s}e^{-\frac{E_{a,s}}{RT}}]$$
where $\bar{y_{A}^{U}}$ and $\bar{y_{A}^{L}}$. are the mean concentrations of *A* in the upper and lower compartment of the diffusion chamber, respectively, provided *A* is applied to the upper compartment; *ψ*, ⟨*r*⟩, *D_s_,_0_*, and *E_a_* are used as fitting parameters when applying the model to the experimental data. 

### 2.4. Gas Permeability 

The pressure-induced gas velocity, *v*, of a porous structure is described by Darcy’s law, which assumes proportionality of gas velocity and pressure gradient: (13)$$v=\frac{Q}{A}=-\frac{K}{\mu }\frac{\partial p}{\partial x}$$where *Q/A* is the volume flow per cross-section, *K* is the permeability coefficient, and *μ* is the gas viscosity. Steady-state conditions allow for integration over height of the permeable layer, *L*, yielding the following expression for experimental determination of *K*: (14)$$K=-\frac{Q}{\Delta p}\frac{L}{A}\mu$$

## 3. Materials and Methods 

### 3.1. Sample Characterization

The samples used for this study were provided by an industrial manufactory of clay bricks. The raw material of sample A consisted of a carbonate-rich clay (~18 w-%), which was blended with organic additives (~4 w-%) and extruded for brick forming. Samples were cut out of a real dried, but unfired, brick. A second sample, B, was tested in the diffusion chamber. This reference material incorporated almost no carbonates, but similar organic and also reasonable metal oxide content. Experiments should clarify the role of carbonates for the CO_2_ propagation, which was observed during heat-up of the carbonate-rich sample A. 

Disc-shaped specimens with a diameter of 88 mm and a thickness of 8 ± 0.1 mm were adapted to the design of the diffusion chamber presented in Section 3.3. Specimens were pretreated in a muffle oven (Nabertherm, Lilienthal, Germany) at different temperatures in order to investigate the influence of the degree of burnout on the transport characteristics of the material. For the preparation, the muffle oven was heated to a constant temperature, and the specimen was pretreated for 2 h. In the case of the 900 °C pretreatment, the sample was put into the oven at a temperature of 550 °C and continuously heate to the maximum temperature before the constant phase, to avoid thermal stresses. 

Table 2 shows X-ray fluorescence spectroscopy (XRF) analysis and total inorganic and organic carbon (TIC, TOC) of the raw clays used for the sample mixtures A and B. The numbers show very similar chemical composition, with the exception of the metal oxide compounds, which is balanced in clay B with Al_2_O_3_ and loss on ignition (LOI) in comparison with clay A. For sample mixtures A and B, organic and inorganic compounds had been added to increase the overall porosity. TIC/TOC; LOI; apparent density, *ρ_app_*, and intrinsic density, *ρ_intr_*; calculated porosity, *ε_calc_* = 1 − *ρ_app/_ρ_intr_*; and increase of porosity, Δ*ε*, by temperature pretreatment of these mixtures are listed in Table 3. Intrinsic densities have been estimated by the raw mixture compositions, containing compounds of different densities. Hg porosity (*Hg ε*), obtained by mercury intrusion porosimetry (Pascal 140 and 440, Thermo Fisher Scientific, Waltham, Massachusetts, USA), was evaluated for a selection of specimens from sample A. Calculated and Hg porosity are quite close to each other, showing both methods to be appropriate for evaluating the porosity of clay mixtures [23]. The values reveal that, for both samples, raw mixture porosity is already substantial and increases by burnout by ~5–15 v-%. It should be noted that, at a preparation temperature of 450 °C, the mass loss of sample A mainly originates from combustion of the organic components of the raw mixture and water loss of the layered silicates in the clay, whereas the LOI step between 450 °C and 900 °C is consistent with the mass of CO_2_ (~8 w-%) emitted by decarbonatization of 18 w-% carbonates. Sample B was characterized by a higher initial apparent density compared to sample A, and a lower initial porosity, provided it had the same assumed intrinsic density. 

Sample A was further characterized by the pore size distribution (from Hg porosimetry) and particle size distribution of the clay foundation, investigated by a laser diffraction counter (Mastersizer 2000, Malvern Instruments, Malvern, UK). Figure 1 shows pore and particle size distributions in the same plot. Pore structure was analyzed for three samples: an extruded clay material of clay foundation A prepared at 600 °C in the muffle oven according to the prescribed procedure, and two sample mixtures prepared at 600 and 900 °C. The modal pore diameter of the pure clay sample after 600 °C burnout was ~0.6 μm; with organics blending, it was slightly higher with ~0.85 μm. Burnout at 900 °C shifted the modal pore diameter to ~1.4 μm, while the size distribution is narrower, indicated by the rise of the relative frequency of the modal value. Overall, the modal pore diameter is around 1 μm, which is consistent with the particle size distribution: Because the smallest detected particles are around 0.4–1 μm and assumed to be spherical, interstices are expected to be in the same order of magnitude and, hence, similar to pore sizes with the highest incidence.

Dondi et al. provided a characterization study of a set of thirteen clay bricks fired at temperatures close to 1000 °C [6]. Porosities were found in the range of ~30 to 40 v-% and mean pore diameters in the range of ~0.4–1 µm. The authors also showed that such clays deployed in coarse-ceramics industry are very similar in other characteristics, such as particle size distribution and mineralogical composition. The characteristics of the clays deployed in the present work are well retrieved within that dataset.

### 3.2. Experimental Set-Up

The apparatus used for the steady-state counter-diffusion experiments, also known as a Wicke–Kallenbach cell, involves two different gas streams applied to the compartments of a diffusion chamber, which are separated by a (porous) sample layer. Via the concentration gradient between the two compartments and the provided isobaric conditions, a diffusive flux across the layer is established, which is detected by a concentration change at the gas outlets. 

A diffusion chamber able to perform experiments at elevated temperatures was designed and manufactured in the laboratory. A schematic cross-section is depicted in Figure 2. The chamber mainly consists of two stainless steel flanges holding in position the disc-shaped specimen, mounted with screws. Welded connections to the inlet and outlet pipes ensure a gas-tight system. Between the flanges and the specimen, high-temperature resistant gaskets are used for sealing (Klinger PSS 200, Gumpoldskirchen, Austria). A steel ring bordering the specimen with a distance of 5 mm serves for absorbing a part of the pressure force induced by the screws as well as separating the diffusion chamber against its surroundings. The sample is kept in position only by the gaskets at both sides. Unlike in many examples in the literature, a sample holder is not used in order to avoid difficulties with thermal dilation or unknown leakages in a sealing between holder and sample. With an overlap between specimen and gaskets of 5 mm, a diameter of 77 mm is in contact with the gas phases of the compartments. The tightness of the seal was tested with a sample dummy constructed out of steel. The steel disc dummy was also deployed to shape the gaskets before use in measurements, in order to minimize mechanical forces on the clay specimen by the mounting. To prevent any pressure drop of the outlet tubes and consequently a pressure difference between the two compartments, the outlet tubes have a larger diameter of 9 mm, compared with the inlet tubes, with an inner diameter of 4 mm. Isobaric conditions are monitored by a pressure transducer (Kalinsky DMU 2, 0–50 Pa, Erfurt, Germany). The steel probes for pressure sampling and thermocouple passage are positioned directly at the compartments. Temperature is measured regularly at both sides of the diffusion chamber. The whole chamber, consisting of the flanges, connected with inlet and outlet tubes, detection ports and screws, is encased within a programmable lab kiln. The gas tubes and ports are passed through holes in the kiln containment. Inlet tubes are connected to a gas supply provided by mass flow controllers, and outlet tubes are led to the off-gas line, while analyzing gas is withdrawn from outlet tubes close to the chamber outlets. The analyzing gas flows to a cooler and then to a non-dispersive infrared gas analyzer (ABB, URAS26, 0–20 v-%, Zürich, Switzerland) for CO_2_ detection and an electro-chemical sensor (GS Yuasa, model KE-25, 0–25 v-%, Kyōto, Japan) for O_2_ detection. In Figure 3, a picture of a prior version of the chamber is shown.

### 3.3. Method of Diffusion Chamber Measurements

In a typical experiment, the mounted diffusion chamber with the specimen was placed in the oven and inlet and outlet tubes and probes were connected to the test rig environment. After mounting, the setup was continuously heated with a rate of 150 K/h. The pressure difference was below 5 Pa for all experiments. Before beginning, the chamber was purged with nitrogen to remove oxygen and other sorbed gases from the chamber compartments and the specimen, which was observed by the measured concentrations reaching the offset levels. Two different modes were used for diffusivity measurements: (i) experiments were performed with pure gases combining O_2_, CO_2_, and N_2_, which were applied to the upper and the lower compartments, respectively; (ii) ternary mixtures were investigated by applying 50 v-% O_2_/N_2_ and CO_2_/N_2_ mixtures to the lower and the upper compartments, respectively (O_2_/N_2_ in counter-current with CO_2_/N_2_). After purging, the gas mixture was applied and breakthrough was monitored subsequently by the change of the off-gas concentrations. Volume flows were 1 sl/min at each compartment for all experiments. Data were obtained in two different manners: (i) fixed gas adjustment over the whole temperature range and sampling of the changing off-gas concentration; (ii) single data points were taken one after another by adjusting the gas mixture after steady-state was reached at each point. This enabled one to conduct several gas mixture profiles over temperature within the same heat-up run. 

Permeability coefficients were obtained by applying a nitrogen gas flow to the upper compartment, while the inlet of the lower compartment was closed (see Figure 2). Needle valves were used to apply a pressure difference between the compartments, while the total flow was divided into two parts. One part was forced across the porous sample and measured at the lower outlet, and the rest crossed the upper compartment and was measured at the upper outlet. Gas flows were detected by gas rotameters. Summing up the two flows allows one to regularly check the pressure tightness of the chamber. Flow/pressure ratios were set in a temperature dependent pressure range from 0 to 1000 Pa for each temperature point, as long as gas tightness was provided.

### 3.4. Leakage Test of Diffusion Chamber

Figure 4 depicts a full temperature range leakage test of the diffusion chamber at ambient pressure. For that purpose, the steel dummy instead of a sample specimen was mounted in the chamber, and the blank test conducted analogously to diffusivity test runs. The chart shows some small but inevitable changes of pressure difference between the lower and upper compartments by adjustment of the gas supply in the range of +/− 5 Pa. For higher detection resolution, CO_2_ was detected by a 5000 ppm infrared cell (SE-0018 K30, Ormond Beach, Florida), only for the blank test. The leakage test was conducted during heat-up by several consecutive phases of O_2_/N_2_ or CO_2_/N_2_ binary mixtures applied to the chamber. The figure shows concentrations of diffusants CO_2_ and O_2_ at the outlet of the compartment opposite the one where the diffusant had been applied. From the figure, one can see a small leakage, resulting in 0.15–0.25 v-% concentration changes for both O_2_ and CO_2_, with a rise in temperature. Because detected diffusive fluxes resulted in concentration changes in the range of 1–15 v-%, this amount of leakage was deemed acceptable, and results from diffusivity tests were adjusted based on the leakage test. Only at temperatures below 100 °C is leakage higher. The deployed sealing material is particularly suitable for elevated temperatures beyond 100 °C. Another material might have been more appropriate for ambient conditions, which should be considered if these conditions are of major interest. However, the phenomenon might have induced higher experimental deviations from actual diffusivity and permeability at ambient conditions than for the rest of the temperature range, although this wasalso considered in the data adjustment.

## 4. Results and Discussion

### 4.1. Temperature-Dependent Molar Fluxes of Diffusants

Most experiments were conducted with sample A, while sample B was used as a reference. Molar fluxes were determined by mass balance of the diffusant across the diffusion layer, considering non-equimolar flow for gas-phase diffusion and unidirectional surface diffusion, which required iteration between calculated experimental flux and gas flux according to the model equation, Equation (12). Figure 5 shows a summary of the binary-mixture experiments over temperature of sample A. The data represent the molar fluxes of the diffusant across the diffusive layer. Curves in the figure indicate continuous measurements, while dots represent single point experiments, as described above. Designation of datasets specify the diffusant and sample pretreatment temperature, which is a measure for burnout of the specimen investigated. The higher the pretreatment temperature, the wider was the temperature range in which measurement was possible without oxidation or calcination reactions interfering with the diffusion process. As volume flows were always the same for both compartments, the molar flux profiles provide an overview of the differences in diffusivity between specimen as well as different gas species.

Two identical measurements of the specimen prepared at 900 °C shown in Figure 5, indicated by #1 and #2, reveal reasonable overall reproducibility with a relative difference of ~10%, which might result from natural inhomogeneity of the samples as well as accumulated precision errors of the test rig. All curves show a moderately declining rise at lower temperatures, whereas behavior differs substantially from about 400 °C between the O_2_ and CO_2_ curves; O_2_ reaches a plateau, while CO_2_ changes to a progressive rise with temperature. Two groups of curves are identified in terms of sample preparation: the tested specimen with a burnout temperature up to 600 °C, and curves from the 900 °C specimen (fully converted). The increase of diffusivity is small for all samples with moderate burnout temperatures compared to the as-received ones (105 °C), but is much higher for the fully converted specimen. As previously mentioned, at temperatures lower and up to 450 °C, only the organic compounds of the raw sample mixture were removed, while complete decarbonatization was achieved by burnout temperatures of 900 °C. Organic content of sample A was substantial, but smaller than CO_2_ bound in carbonates (~4 w-% organic burnout vs. ~8 w-% decarbonatization weight loss). Apparent porosity differed from the as received samples by 3.4 v-% for organics burnout and by a further 5.2 v-% for decarbonatization (see Table 3). Due to the lower intrinsic density of the organic compounds, which was estimated at ~790 kg/m³ compared to the carbonates with a density of about 2700 kg/m³, the rise of apparent porosity in relation to the mass loss is larger for organics compared to inorganics. However, diffusivity seems not to increase proportionally, but is stronger by decarbonatization. Hence, it appears that decarbonatization seems to promote diffusivity to a greater extent than the removal of the organics.

In Figure 6, a comparison of binary-mixture and ternary-mixture measurements with a sample A specimen is presented. As mentioned before, O_2_/CO_2_ were diffusing in counter-current with N_2_ in binary-mixture measurements, while in the case of ternary mixtures, O_2_/N_2_ was in counter-current with CO_2_/N_2_. It can be readily seen that diffusive fluxes were roughly double when the inlet concentration was 100% (binary diffusion), compared to the 50% pre-mixtures (ternary mixtures), regardless of the counter-current species. From a temperature of about 400 °C, a progressive rise of CO_2_ flux was again observed, diverging from the prior proceeding. Analogously, O_2_ curves seem to reach a plateau, which is not explained by gas-phase diffusion theory. Possibly, the temperature-activated inner surface of the porous media prevented O_2_ from unhindered diffusion.

Sample B showed the same temperature-dependent behavior as sample A, whereas the main difference between the two samples was the carbonate content (18 w-% for A and almost none for B) and the metal oxide content of the as-received material (8.8 w-% vs. 1.9 w-%). Figure 7a depicts the O_2_ and CO_2_ diffusivity of samples A and B, revealing generally lower diffusivity of sample B, but also a massive CO_2_ diffusion rise with temperature for both samples, analogously to the leveling of O_2_. Diffusivity rise with burnout degree is shown in Figure 7b for samples A and B. The data pairs represent measurements of O_2_ or CO_2_ diffusivity, conducted at ambient temperature with the as-received compared to the fully converted samples. As can be seen, diffusivity of A and B was similar for the as-received specimen, but enhancement by burnout was substantially different. This indicates that the calcination of the carbonates of sample A was responsible for the pronounced diffusivity rise. 

### 4.2. Diffusive Transport Characteristics of Investigated Samples

The modelling of O_2_ diffusion was achieved with gas-phase diffusion theory, but was limited to a temperature range from ambient to 500 °C due to an observed, but not fully explainable, leveling of O_2_ diffusive flux at higher temperatures. The pressure difference between the two compartments of the diffusion chamber was kept at ~zero, in order to observe only diffusive transport and to exclude convective flux, i.e., permeation. The extended diffusion model for gas-phase and surface diffusion according to Equation (12) has been applied to the experimental data by least-squares fit (in the case of O_2_, the second term of the right-hand side of Equation (12), which considers surface diffusion, was assumed to be zero). Although rigidly correct only for binary gas diffusion, Equation (12) was also deployed to the ternary mixture experiments. In that case, the molecular diffusion coefficients and the molar masses of the counter-current species were averaged (e.g., O_2_ diffusing into a mixture of CO_2_ and N_2_). However, the binary model should not be used for ternary mixtures if the molar masses of the gas species (and thus the molecular diffusion coefficients) differ to a much greater extent than in the present set of gas species. The model was applied to each dataset of identical samples (sort and preparation) comprising different gas system experiments in order to derive species-independent transport characteristics. Figure 8 depicts an exemplary data fit for sample A with a burnout temperature of 900 °C. Dots, crosses, etc. show experimental data and curves show the model predictions after fitting. The relative fitting residuals were typically below 10%, and often below 5%, within the defined fitting temperature ranges.

In Table 4, the resulting transport characteristics are listed. The mean transport-pore radii, *<r>*, were evaluated with numbers around 0.5 μm, therefore well agreeing with the modal pore diameters derived from Hg porosimetry (see Section 3.1). It is worth mentioning that start guesses of *<r>* for the fitting procedure were set to 1 μm, which might have induced finding the local minimum that best matched the experimental data. The parameter *ψ* rises with the burnout degree of sample A, particularly after complete conversion with pretreatment at 900 °C. Value of sample A with 450 °C pretreatment is equal to sample B with 900 °C pretreatment. This is remarkable because one can assume full conversion of organic compounds of sample A at 450 °C, whereas carbonates are unconverted, and sample B does not contain carbonates. Hence, these two samples are comparable not only in *ψ*, but also in the porous structure. Oscarson found very similar *ψ* numbers in the range 0.034–0.093 when letting cations diffuse in a water solution within different clays [22]. The results for sample A with a preparation temperature of 600 °C are not consistent with the rest of the dataset, indicating that only one tested gaseous system might not be sufficient for species-independent evaluation of transport characteristics. 

Further, the results of the model application show that the generally lower diffusivity of sample B is described mainly by its smaller porosity, resulting in a smaller *ψ* number. The model therefore appears to give a sound mechanistic explanation of the actual diffusion process in porous media. *D_s_^0^* and *E_a,s_* for CO_2_ surface diffusion of both fully converted clay samples (900 °C) are similar; the numbers found for the raw or partially converted specimen differ from those values. However, they may have reduced relevance due to the fact that the actual experimental temperature ranges were limited in order to avoid interference of the signal by occurring chemical reactions (see Figure 5). Horiuchi et al. reported on the heat of adsorption of CO_2_ on alumina modified with metal oxides such as CaO, MgO, etc. [8]. They found values around 150 kJ/mol, about three times higher than the apparent activation energy of the present work. This is in line with findings of other researchers reporting a ratio between heat of adsorption and activation energy for surface diffusion of around 2.5 [19]. The activation energy may exhibit a dependency of surface coverage of the pore walls by the diffusant. Hence, the results probably would have been different for supply concentrations of diffusants other than those applied in the present experiments. 

### 4.3. Permeability 

Permeability, *K*, was evaluated for sample A with different temperature pretreatments with numbers in the range of 1 ∙ 10^−13^–2 ∙ 10^−11^ m², covering two orders of magnitude over the investigated temperature range. The burnout degree was shown to exhibit no significant influence on the apparent permeability, which can be seen from Figure 9. Error bars indicate the standard deviation obtained by several pressure/flow ratios, as discussed in the methods section. The curve in Figure 9a shows a simple polynomial fit, able to serve as a temperature dependent material property in a momentum balance. However, an Arrhenius plot of the data points reveals an exponential rise with temperatures above 200 °C, Figure 9b. Deviation of that proceeding at lower temperatures might have resulted from lower detection limits.

Weller et al. provided a literature overview of the permeability data of sandstones [24]. Comparison shows that permeability of ~1 ∙ 10^−13^ m², determined at ambient temperature in the present study, is in good agreement with sandstone permeability reported by many researchers. Additionally, other authors have reported data for ceramic materials that are consistent with the present findings, particularly if the mean pore diameter is considered [25,26]. Sandstone permeability was reported to exhibit progressive temperature dependency [27]. Hence, the permeability/temperature relationship presented here seems to correctly reflect the actual proceedings.

### 4.4. Discussion

Figure 10a shows mean free path length, *λ*, and the Knudsen number for nitrogen, and a mean pore diameter of 0.45 μm (corresponding to sample A, prep. temp. 900 °C, Table 4), according to Equations (2) and (3) over temperature. Mean free path lengths of relevant gas species and mean pore diameters derived from the Hg porosimetry measurements of clay samples (Figure 1) are in the same order of magnitude, resulting in Knudsen numbers ~1. Hence, Knudsen diffusion is theoretically relevant over the whole range of temperatures. Figure 10b depicts effective molecular, effective Knudsen, and combined effective gas-phase diffusion coefficients over temperature for sample A (prep. temp. 900 °C), which were derived by fitting the extended diffusion model given Equation (12) to the experimental data. It transpires that the overall diffusivity is affected by both mechanisms, but is closer to the effective molecular diffusivity. The influence of the Knudsen mechanism is moderate, reflected by *Kn* numbers spanning from 0.05 to 0.20 in the concerned temperature range, see Figure 10a. It is obvious that the rising trend of *Kn* with temperature corresponds with the increasing relevance of the Knudsen mechanism to the overall diffusivity. 

Peng et al. found a diffusion coefficient of 0.014 cm²/s for a clay brick [4], which is very close to the model prediction represented by Figure 10b, revealing an apparent gas-phase diffusion coefficient at an ambient temperature for sample A of 0.011 cm²/s.

Molecular diffusion basically scales with T^(3/2), while the temperature dependency of Knudsen diffusion is T^(1/2). Temperature dependency of effective diffusion coefficients is frequently assumed to be exponential and described by an Arrhenius equation (e.g. [5,28]). The present results in Figure 10 show that, in porous media, the actual diffusion mechanism can differ from a progressive rise in temperature by the dominant influence of the Knudsen diffusion regime. The *Kn* number provides a good estimation of Knudsen mechanism relevance, which is small if *Kn* ≲ 0.1.

At moderate temperatures below ~400 °C, the experimental data exactly fulfill non-equimolar diffusive flow according to Graham’s law, so that O_2_ fluxes were generally higher than CO_2_ fluxes (see Figure 5, Figure 6 and Figure 7). Applying Equation (8), negligible residuals were obtained frequently, which supports the validity of the experimental method. Table 5 depicts some fluxes measured during investigations for fully converted sample A. 

From temperatures around 400 °C, the O_2_/CO_2_ flux relation began to switch and CO_2_ diffusivity increased strongly with temperature, whereas O_2_ flux stagnated at a certain level or even slightly decreased. Possible chemical reactions acting as CO_2_ sources or O_2_ sinks seem unlikely to be responsible for that behavior. The substantial carbonate content of sample A gave rise to the assumption that this could be related to the strong observed surface diffusion effect, which could also prevent O_2_ from unhindered diffusion at the same time. The activation of carbonates from temperatures beyond ~400 °C would fulfill the role of active sites, promoting the surface transport of CO_2_. However, sample B clay with substantially lower metal oxide concentrations showed the same behavior, indicating that metal oxides are not responsible for the CO_2_ propagation. No concentration dependency of the effect was found, considering the difference in metal oxide concentrations of 8.8 w-% vs. 1.9 w-% for clay A and B, respectively (see Table 2). Still, metal oxides could be in excess for CO_2_ interaction in both samples. However, it is likely that other clay compounds promote the CO_2_ diffusion. Clays typically contain substantial amounts of alumina (Al_2_O_3_) with portions around 15–25 m-% (Table 2 and [23]). Rivarola and Smith tested CO_2_ diffusing in alumina pellets at room temperature, reporting that besides gas-phase diffusion (molecular and/or Knudsen), surface transport can substantially contribute to diffusion fluxes, [21]. Zettlemoyer et al. reported a successful deposition of polarizable liquids on several clay compounds [29]. Zeolites, which comprise similar compositions to clay minerals, are reported to be promising candidates as CO_2_ adsorbent materials [30]. CO_2_ adsorption in that context is described as an interaction of the gas molecules with the electrostatic field acting in phyllosilicates between aluminum atoms and exchangeable cations such as Na and K. The cations compensate the negative layer charges generated by aluminum atoms in the case of AlO_4_ tetrahedra, in comparison to SiO_4_ (with 3 and 4 valence electrons of Al and Si, respectively). Hence, aluminum content of the clay might be a measure for the strength of the surface diffusion effect. CO_2_ exhibits a ten-times-higher quadrupole moment compared to O_2_ (see Table 1), which is able to interact with electro-static field gradients. Thus, CO_2_ surface diffusion might be promoted by the same effect as the adsorption of CO_2_ or other polar species in alumina, while O_2_ was not affected due to its weak interaction with electrostatic fields.

However, the finding of there being a progressive rise of diffusivity is contrary to that of other authors, who describe surface diffusion as exhibiting decreasing temperature dependency due to lower adsorption rates, which let surface transport decrease [19,31]. This discrepancy should be addressed theoretically and empirically in further investigations to clarify the observed phenomenon.

## 5. Conclusions

In the present work, the temperature dependent diffusivity and permeability characteristics of two exemplary clay materials used for brick manufacturing were investigated. Results were modelled with the mean transport-pore model (diffusion) and Darcy‘s law (permeation). The most remarkable findings were as follows:The determined diffusion and permeability coefficients at ambient conditions were ~1 ∙ 10^−6^ m²/s and ~1 ∙ 10^−13^ m², respectively, in line with other literature. However, for the temperature dependency, comparable data is scarce at the present time.The variation of the pretreatment temperature affected the diffusivity of the clay material. Burnout of organic and inorganic compounds of the clay mixtures increased porosity, which was followed by a substantial rise of diffusivity. Results indicated that calcination of incorporated carbonates had a greater effect on the diffusivity than the burnout of organic compounds.At temperatures beyond ~400 °C, CO_2_ diffusivity increased progressively with temperature, indicating a strong selective surface diffusion effect.Permeability was not affected by the burnout degree and showed progressive (probably exponential) temperature dependency.

As has been shown, diffusivity can change during temperature exposition, following growth of the pore volume. The results further revealed that temperature dependency is moderate unless surface diffusion comes into play. Experimental data showed excellent agreement with Graham’s law at a moderate temperature range up to ~400 °C, resulting in a higher molar diffusivity for lighter gases. Diffusive transport parameter, *ψ*, spanned from 0.012 to 0.056, and mean transport-pores, *<r>,* were around 0.5 µm. The characteristic *ψ* showed a rising dependency of the burnout degree relative to the porosity. Values were equal for the two investigated clay samples for comparable sample conditions. However, the parameter increased by a factor of ~2 after decarbonatization of the carbonate-rich sample. Consideration of CO_2_ surface diffusion required expansion of the model, yielding pre-exponential factors, *D_s_^0^*, and activation energies, *E_a,s_*. Similar numbers have been assessed for both fully converted clay samples (900 °C) with orders of magnitude of 5 ∙ 10^−3^ m²/s and 50 kJ/mol, respectively. Thus, the phenomenon of CO_2_ surface diffusion seem to be a characteristic of natural clays at temperatures beyond ~400 °C. The variety of clays and experimental methodology should be extended in further investigations in order to substantiate that hypothesis.

## Figures and Tables

**Figure 1 materials-14-04942-f001:**
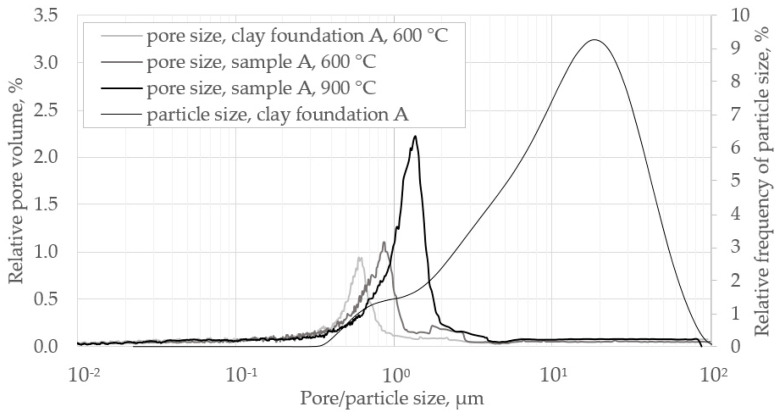
Pore size distributions of prepared sample A and particle size distribution of clay foundation A.

**Figure 2 materials-14-04942-f002:**
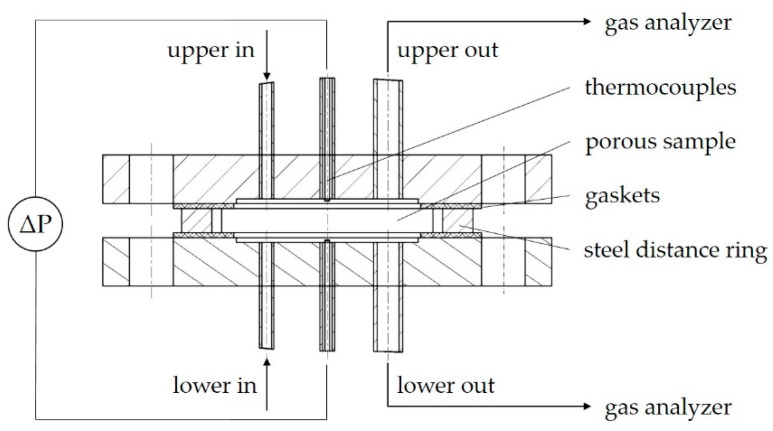
Cross-section of the diffusion chamber integrated in the diffusion test rig.

**Figure 3 materials-14-04942-f003:**
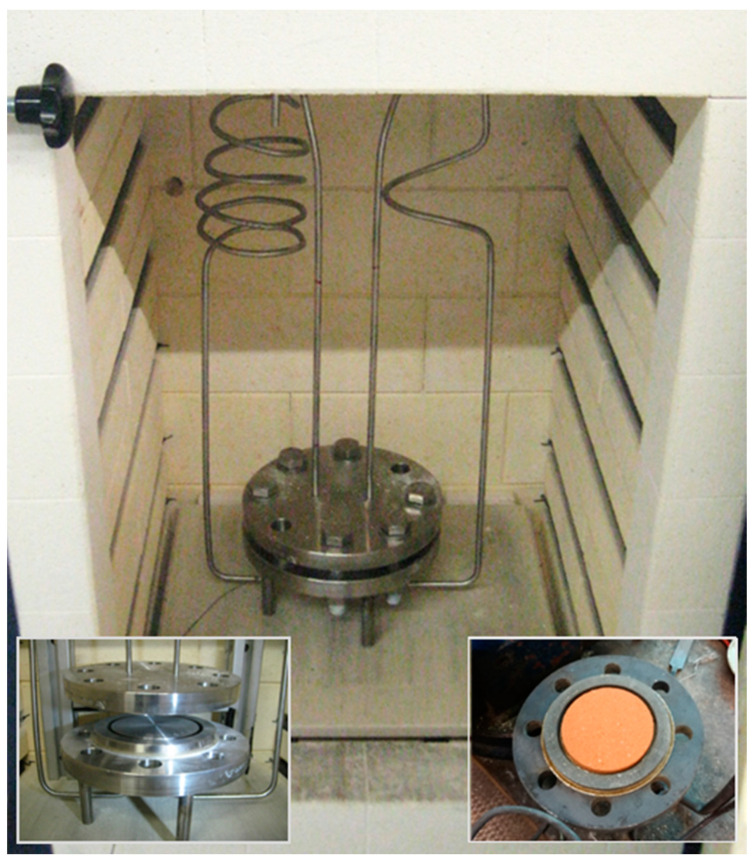
A prior version of the diffusion chamber mounted to the programmable lab kiln; insets show the opened chamber with spacer ring and steel dummy (left) or clay specimen (right).

**Figure 4 materials-14-04942-f004:**
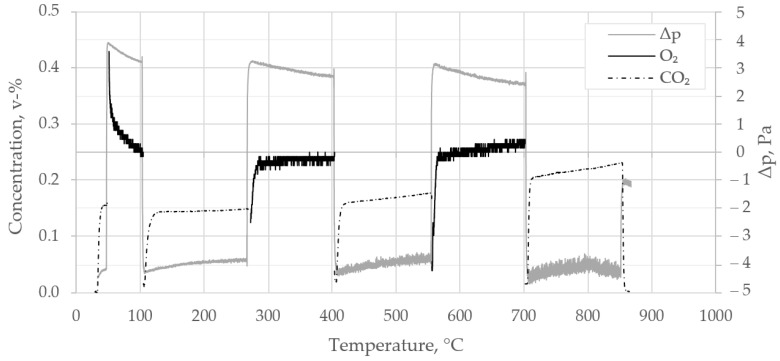
Leakage test of diffusion chamber with applied gas combinations O_2_/N_2_ and CO_2_/N_2_ over temperature.

**Figure 5 materials-14-04942-f005:**
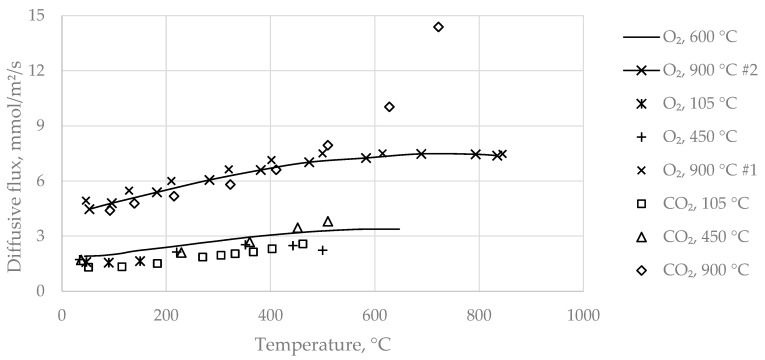
Overview of sample A binary-mixture measurements over temperature; experiments represented by curves were carried out by continuous measurement and dots as single points.

**Figure 6 materials-14-04942-f006:**
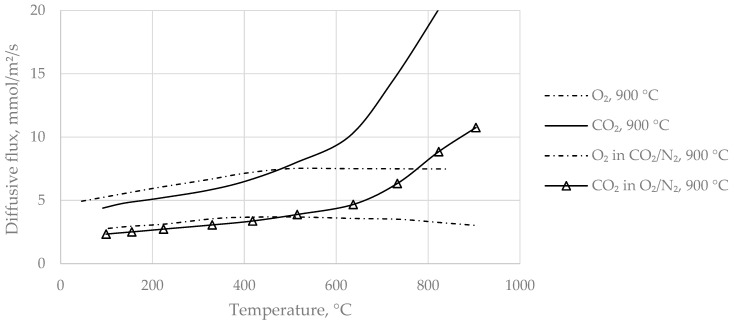
Comparison of binary-mixture (O_2_, 900 °C and CO_2_, 900 °C) and ternary-mixture experiments of sample A.

**Figure 7 materials-14-04942-f007:**
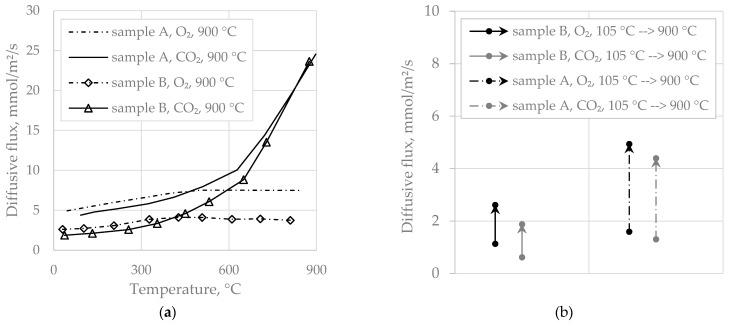
(**a**) Comparison of O_2_ and CO_2_ diffusivity of samples A and B; (**b**) O_2_ and CO_2_ diffusivity rise (at ambient temperature) by burnout of sample A and B specimens.

**Figure 8 materials-14-04942-f008:**
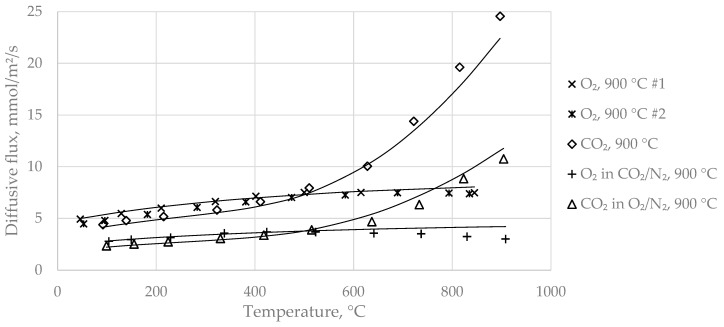
Data fit according to Equation (12) for sample A and preparation temperature 900 °C with gaseous systems O_2_/N_2_, CO_2_/N_2_, and O_2_/CO_2_/N_2_. Dots are experimental data and lines are the model predictions.

**Figure 9 materials-14-04942-f009:**
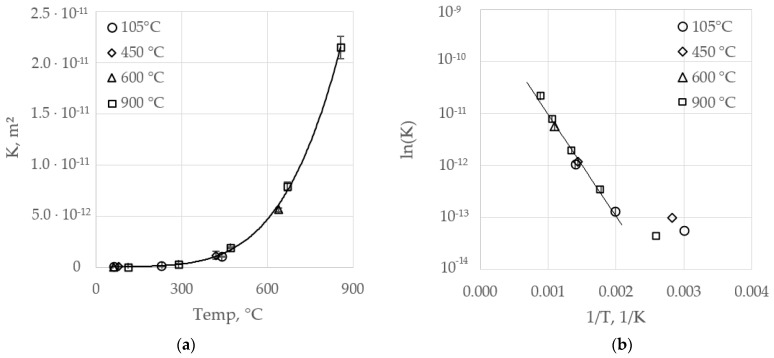
Permeability over temperature of several specimens from sample A; data designation indicates the pretreatment temperature; (**a**) polynomial fit; (**b**) Arrhenius plot of permeability data (three points at lower temperatures excluded from regression).

**Figure 10 materials-14-04942-f010:**
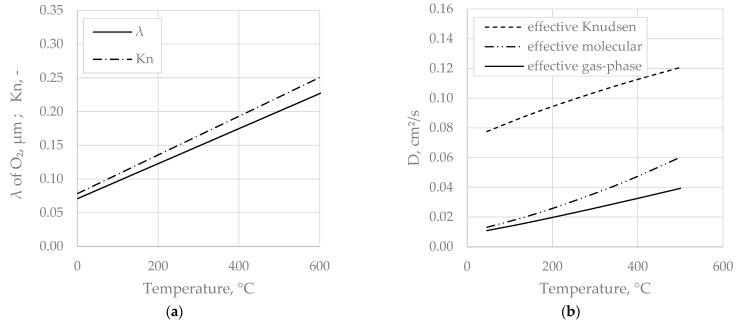
(**a**) Mean free path length, λ, and Knudsen number for N_2_ over temperature; (**b**) effective molecular, Knudsen and combined gas-phase diffusion coefficients for sample A.

**Table 1 materials-14-04942-t001:** Kinetic diameter, *σ*, and quadrupole moments, *Θ*, of relevant gas species, taken from Gaskell [15,17], respectively.

Gas	*σ*, Å	*Θ*, 10^−26^ esu
N_2_	3.681	−1.4
O_2_	3.433	−0.4
CO_2_	3.996	−4.3

**Table 2 materials-14-04942-t002:** XRF chemical composition of the raw clay foundations A and B, in m-%.

Clay	SiO_2_	Al_2_O_3_	TiO_2_	Fe_2_O_3_	CaO	MgO	K_2_O	Na_2_O	SO_3_	LOI	TIC ^1^	TOC
A	53.1	16.0	0.8	6.3	5.0	3.8	2.9	0.7	0.79	11.2	14.4	0.49
B	55.0	24.0	1.2	7.4	0.7	1.2	2.5	0.8	0.04	6.8	0.20	0.33

^1^ TIC as CaCO_3_.

**Table 3 materials-14-04942-t003:** Characteristics of investigated clay mixture samples A and B.

Sample	Sample Preparation	TIC ^1^, w-%	TOC, w-%	LOI, w-%	*ρ_app_*, kg/dm³	*ρ_intr_*, kg/dm³	*ε_calc_*^2^, v-%	*Hg ε*, v-%	Δ*ε*, v-%
A	- ^3^	~18	~2	-	1.72	2.47	30.4	n.a.	-
A	450 °C	-	-	5.1	1.63	2.54	35.8	n.a.	5.4
A	600 °C	-	-	7.5	1.60	2.68	40.3	40.8	9.9
A	900 °C	-	-	13.1	1.49	2.68	44.4	43.1	14.0
B	- ^3^	~0.2	~2	-	1.82	2.47	26.3	n.a.	-
B	900 °C	-	-	8.3	1.67	2.68	37.7	n.a.	11.4

^1^ TIC as CaCO_3._
^2^ ε_calc_ = 1−ρ_app/_ρ_intr._
^3^ as received (dried at 105 °C).

**Table 4 materials-14-04942-t004:** Transport characteristics of investigated samples.

Sample		A	A	A	A	B
pretreatment temp, °C		105	450	600	900	900
gas systems tested		O_2_/N_2_ CO_2_/N_2_ O_2_/CO_2_/N_2_	O_2_/N_2_ CO_2_/N_2_ O_2_/CO_2_/N_2_	O_2_/N_2_	O_2_/N_2_ (2x) CO_2_/N_2_ O_2_/CO_2_/N_2_	O_2_/N_2_ CO_2_/N_2_ O_2_/CO_2_
Gas-phase diffusion	*ψ, -*	0.012	0.025	0.018	0.056	0.024
	*<r>*, μm	0.51	0.23	3.73	0.45	0.62
CO_2_ surface diffusion	*D_s_^0^*, m²/s	9.1 ∙ 10^−6^	2.0 ∙ 10^−4^	n.a.	5.2 ∙ 10^−3^	3.7 ∙ 10^−3^
	*E_a,s_*, kJ/mol	18	35	n.a.	57	52

**Table 5 materials-14-04942-t005:** Diffusive fluxes of O_2_/N_2_ and CO_2_/N_2_ binary mixture tests with sample A (pretreatment 900 °C) and comparison with Graham’s law (see. Equation (8)).

Temp, °C	*N_O_**_*₂*_*, mmol/m²/s	*N_CO_**_*₂*_*, mmol/m²/s	*N_O_* *_*₂*_/N_CO_* *_*₂*_*	*(M_CO_* *_*₂*_/M_O_* *_*₂*_)^1/2^*	Residual
139	5.85	5.01	1.17	1.17	0.4%
215	6.37	5.40	1.18	_“_	−0.6%
323	7.01	6.04	1.16	_“_	1.1%
411	7.51	6.84	1.10	_“_	6.3%

## Data Availability

Not applicable.

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
