# Peer review of "Considerations on Temperature Dependent Effective Diffusion and Permeability of Natural Clays"

_materials, 2021, doi:10.3390/ma14174942_

Round 1

Reviewer 1 Report

This paper presents the experimental results of the investigation the diffusion ability and permeability of clay bricks in relation to gas mixtures of O2, CO2 and N2 during the heating to 900 °C. To explain the experimental results, the gas diffusion model was expanded by including surface diffusion in the model equation. The data interpretations seem to have been done well,  as well as the mechanisms of the diffusion capacity dependence’s with temperature. Thus, the article adds valuable new information concerning our knowledge on the identification and demonstration of new approaches to the study of the diffusion capacity at different temperatures and different composition of the initial clay bricks. Thus, I can recommend an article for publication in the Minerals journal, but with minor changes.

Comments.

A gradual increase in the diffusion capacity with temperature may be due to an increase in porosity during decarbonatization and dehydration reactions. This can be confirmed by studying the mineral composition of the starting material and the reaction products formed at different temperatures. But, in my opinion, this is the task of further investigations.

Line 485-486. CO2 adsorption in that context is described as an interaction of the gas molecules with the electro-static field generated by the charge gradient between negative aluminum atoms (???) and exchangeable cations like Na, K, which are also present in natural clays.

Author Response

Dear Sir or Madam,

thank you for your kind attention to our manuscript. Among all revisions carried out, the major ones are:

  • The introduction has been reworked introducing a paragraph which deals with some fundamentals on clay and clay minerals, further the order of text passages was modified
  • The theory section has been advanced with a mechanistic explanation of the mathematics of the diffusion model equations
  • In the sample characterization part as well as in the results section, some references has been added, underlining the accordance of the present results with literature
  • A picture of the diffusion test rig has been introduced
  • Unfortunately, the former figure 9, which is figure 10 now, incorporated a mistake we had to correct. Statements concerning that chart were modified in order to fit the result of model predictions shown in figure 10. As a consequence, the whole data processing was checked once more, without finding any further errors. We regret that issue.
  • Concluding remarks have been completely reworked focussing on the most considerable points of the work

Reviewer 2 Report

The authors presented a very well done experiment. However, the work requires a slight correction of details, which are presented below:

1) Why the sample B was not measured by Hg porosimetry? 

2) Why the sample B was not included as a compere on figure 1 and 3?

3) On figure 3 the concentration of O2 is decreasing between temperature 0-100C, but between temp 300-400C is almost constans but between temp 600-700C is increasing, please describe this phenomena. 

Author Response

Dear Sir or Madam,

thank you for your kind attention to our manuscript. Among all revisions carried out, the major ones are:

  • The introduction has been reworked introducing a paragraph which deals with some fundamentals on clay and clay minerals, further the order of text passages was modified
  • The theory section has been advanced with a mechanistic explanation of the mathematics of the diffusion model equations
  • In the sample characterization part as well as in the results section, some references has been added, underlining the accordance of the present results with literature
  • A picture of the diffusion test rig has been introduced
  • Unfortunately, the former figure 9, which is figure 10 now, incorporated a mistake we had to correct. Statements concerning that chart were modified in order to fit the result of model predictions shown in figure 10. As a consequence, the whole data processing was checked once more, without finding any further errors. We regret that issue.
  • Concluding remarks have been completely reworked focussing on the most considerable points of the work

Please also see the attachment.

Reviewer 3 Report

The presented paper is interesting and very useful for different scientists from different fields –chemical engineering,  geology, environment protection. Much data are gathered and re-processes to reach the conclusions related to dependent of effctive diffusion and permeability of natural clays. The research of this article mainly focuses in clays has certain practical significance. However, the paper should be once more carefully read in to correct some technical omissions. Examples point to the point:

  • the intoduction is enough poor of references in total. Please add more references about the natural clay minerals. 
  • materials and methods in my opinion is enough poor for manuscript in materials and without any picture for unknown scientists.
  • experimental procedure. In my opinion we must change the title.
  • in my opinion it is obligatory all abbreviations to be introduced when they appear for the first time, PERHAPS in the intrduction part.
  •   318-329 lines . Unclear, please revise the expression.
  • The discussion 4.4 is very comprehensive
  • I prefer conclusions with bullets because your conclusions are very comfused
  • biblography: please add more references, only 25 references are not enough. 

Author Response

(The authors gave the same response as above.)

Round 2

Reviewer 3 Report

Accept in present form

Author Response

Thank you for your efforts in reviewing our manuscript! English improvements may follow in the coming editorial process, according to a message of the responsible editor.

Kind regards, F.W.
